# Structural determinants for GPCR-mediated inhibition of TASK K2P channels by diacylglycerol and its dysfunction in disease

Thibault R H Jouen-Tachoire [1,2,3,4,8], Peter Proks [1,2,8], David Seiferth [5,7], Kate Crowther [1,2], Philip C Biggin [5], Thomas Baukrowitz [6], Marcus Schewe [6✉] & Stephen J Tucker [1,2,4✉]

## Abstract

**Two-Pore Domain K+ (K2P) channels are crucial determinants of the resting membrane potential and of cellular electrical excitability in many different cell types. TASK-1 and TASK-3 K2P channel activity is also coupled to GPCR signalling pathways via Gαq and their subsequent inhibition is via direct interaction with diacylglycerol (DAG) generated from phosphatidylinositol-4,5-bisphosphate (PIP$_2$) hydrolysis. This regulation is defective in two different neurodevelopmental disorders, but the molecular mechanisms underlying this inhibitory process and the reasons for the GPCR-insensitivity of these disease-causing mutations remain unclear. Here we show that GqPCR inhibition inversely correlates with channel open probability, and results from a state-dependent destabilisation of the open state by DAG promoting channel closure. We also identify a DAG interaction-site within a groove between the M2, M3 and M4 domains, and show the crucial role of residues within this site in mediating the inhibitory effect and defining channel sensitivity. These results not only reveal the structural and molecular mechanisms underlying GqPCR regulation of TASK channels, but also explain the pathogenic effect of a common regulatory defect linked to different K2P channelopathies.**

**Keywords** Channelopathy; Diacylglycerol; GPCR/K2P Channel; KCNK3
**Subject Categories** Molecular Biology of Disease; Structural Biology

## Introduction

Potassium (K+) channels are transmembrane pores that allow the selective movement of K+ across cellular membranes to control cellular electrical activity. In particular, the family of two-pore domain K+ (K2P) channels are important for the generation and regulation of the resting membrane potential (Enyedi and Czirjak,

2010). However, although they were originally described as 'leak' channels, they are now known to be regulated by diverse physico-chemical stimuli, as well as a range of GPCR-dependent and other cellular and metabolic signalling pathways (Niemeyer et al, 2016).

There are 15 different human K2P channels encoded by the *KCNK1-18* genes and are grouped according to both their sequence similarity and functional properties. In particular, TASK K2P channels include two members, TASK-1 (*KCNK3*) and TASK-3 (*KCNK9*), which were originally characterised by their sensitivity to inhibition by external protons (H+) across the physiological pH range, a property from which they derive their name (TWIK related acid-sensitive K+ channels) (Sepulveda et al, 2015). GPCR-coupled signalling has also been shown to inhibit TASK channels, and provides a pathway for neurotransmitters and hormones to influence cellular excitability (Mathie, 2007). This GPCR-modulation of TASK currents also plays an important role in the control of aldosterone secretion and a variety of pathways involved in the mechanical control of ventilation, as well as both central and peripheral chemosensation (Bandulik et al, 2015; Bayliss et al, 2015; Trapp et al, 2008). TASK channels are also found in certain nociceptive sensory neurons, as well as the heart, and their crucial role in regulating electrical activity in these various tissues means they have also been proposed as targets for the treatment of pain, atrial fibrillation and sleep apnoea (Kiper et al, 2015; Liao et al, 2019; Mathie et al, 2021).

The inhibition of TASK channels following activation of Gαq-coupled receptors (Inoue et al, 2020; Talley and Bayliss, 2002; Talley et al, 2000) has been suggested to involve several different mechanisms (Mathie, 2007). However, an elegant dissection of this pathway demonstrated a critical role for activation of Phospholipase C (PLC), which triggers hydrolysis of PIP$_2$ into soluble inositol-1,4,5-trisphosphate (IP$_3$) and membrane-bound diacylglycerol (DAG). PIP$_2$ is a known activator of many different K+ channels, but it is not the depletion of PIP$_2$ per se which leads to a loss of channel activity; instead, it is the consequent increase in DAG which leads to a direct inhibition of TASK currents (Wilke et al, 2014). However, the biophysical, kinetic and molecular mechanisms that underlie this direct interaction and ultimate

[1]Kavli Institute for Nanoscience Discovery, University of Oxford, Oxford, UK. [2]Clarendon Laboratory, Department of Physics, University of Oxford, Oxford, UK. [3]Department of Pharmacology, University of Oxford, Oxford, UK. [4]OXION Initiative in Ion Channels and Disease, University of Oxford, Oxford, UK. [5]Department of Biochemistry, University of Oxford, Oxford, UK. [6]Institute of Physiology, Kiel University, Kiel, Germany. [7]Present address: Clarendon Laboratory, Department of Physics, University of Oxford, Oxford, UK. [8]These authors contributed equally: Thibault R H Jouen-Tachoire, Peter Proks. ✉E-mail: m.schewe@physiologie.uni-kiel.de; stephen.tucker@physics.ox.ac.uk

modulation of channel gating remain unknown. TASK channels lack the classical cysteine-rich C1 domains that typically bind DAG, and DAG lacks the more complex headgroup and negative charges that define many phospholipid/channel interactions, especially as seen with $PIP_2$.

Recently we identified a novel TASK-1 channelopathy, Developmental Delay with Sleep Apnoea (DDSA) associated with a range of de novo gain-of-function (GoF) heterozygous missense mutations in *KCNK3* (Sormann et al, 2022). These GoF mutations cluster near the X-gate, a gating motif which regulates intracellular access to the inner cavity and which has previously been implicated in channel activation by volatile anaesthetics, as well as GqPCR regulation (Lin et al, 2025; Luethy et al, 2017). We have previously shown that these DDSA mutations increase channel open probability ($P_o$) and that their GoF effect is exacerbated by a markedly reduced sensitivity to GqPCR inhibition due to a concomitant reduction in sensitivity to inhibition by DAG. Interestingly, similar functional effects are also seen with a number of GoF mutations in TASK-3 associated with another neurodevelopmental disorder, *KCNK9* Imprinting syndrome (KIS) (Barel et al, 2008; Cousin et al, 2022). This therefore suggests that there may be a common gating mechanism that is defective in all these disease-causing mutations in TASK channels, which may provide insight into the molecular basis of their regulation by DAG and GPCR-dependent regulation. Recent structural studies of TASK channels also now provide a wealth of structural information to support such analysis (Hall et al, 2025; Lin et al, 2024; Rodstrom et al, 2020).

In this study, we have used both macroscopic and single-channel recordings of TASK channels to examine their GPCR-mediated inhibition and demonstrate a state-dependent mechanism of regulation by DAG. We also examine the molecular determinants of this DAG-TASK channel interaction and identify a specific site that determines the difference in GPCR sensitivity between TASK-1 and TASK-3. These results not only advance our understanding of the molecular mechanisms underlying GPCR regulation of TASK channels but also the pathogenic effect of TASK channelopathies linked to channel dysregulation.

# Results

## Mutant TASK channels with a range of intermediate open probabilities

We previously reported that all of the DDSA mutant channels identified so far had a markedly reduced GqPCR sensitivity with relatively little difference between them, i.e., there were none with an 'intermediate' or only minor reduction in GqPCR sensitivity (Sormann et al, 2022). Likewise, they all exhibited a marked increase in single-channel $P_o$ of at least tenfold. The analysis of single-channel behaviour can often provide structural and mechanistic insights into ligand action. One possible mechanism that might therefore explain this correlation is a state-dependent mechanism of action of DAG on TASK channels whereby the ligand preferentially alters properties of either the closed or open state to exert its functional effect, similar to the effects seen in other ion channels (Paganelli and Popescu, 2015; Posson et al, 2013; Proks et al, 2021). However, to examine this more fully requires the identification of additional mutant channels with a wider range of

$P_o$ to enable comparison with their GqPCR inhibition and sensitivity to DAG. Our initial screen, therefore, focused on introducing a variety of different amino acid substitutions at three of the key sites adjacent to the X-gate that are mutated in DDSA (F125S, Q126E and G129S) to produce channels with a relatively high $P_o$ and markedly reduced GqPCR sensitivity (Crowther, 2025; Sormann et al, 2022).

Initial two-electrode voltage-clamp (TEVC) recordings show that macroscopic, whole-cell current levels for many substitutions at these positions exhibited an increase in whole-cell activity to a range of different levels (Fig. 1A). We therefore next examined their single-channel behaviour and found that they also exhibited a suitable range of open probabilities, such as those seen with different substitutions at position G129 (Fig. 1B and Table EV1).

## An inverse correlation between GqPCR sensitivity and $P_o$

We next measured mutant channel sensitivity to GqPCR inhibition via co-expression with the P2Y2 receptor (Czirjak et al, 2001; Sormann et al, 2022; Srisomboon et al, 2018). Importantly, we found that those mutations which exhibited a single-channel $P_o$ between 0.01 and 0.7 also exhibited a corresponding range of sensitivities to GqPCR-mediated inhibition (Fig. 1C,D). To better understand this relationship, we plotted these $P_o$ values and observed a clear correlation where GqPCR sensitivity decreases sharply as channel $P_o$ increases. This inverse relationship is most clearly visualised in the plot shown in Fig. 1E.

## Dynamic modulation of channel activity alters GqPCR sensitivity

The mutations examined above can have a variety of functional effects, not just on overall channel $P_o$. A better method is therefore to dynamically modulate the activity of WT TASK channels and measure the effect of these changes on GqPCR sensitivity. Interestingly, both TASK-1 and TASK-3 have been shown to be activated by volatile anaesthetics, and a previous study of TASK-3 reported that its GqPCR sensitivity is reduced upon activation by halothane (Luethy et al, 2017) or chemical modification (Conway and Cotten, 2012). We therefore examined the effect of other mechanisms of channel activation and inhibition on GqPCR sensitivity without the need for mutagenesis.

TASK-1 is known to be inhibited by extracellular $H^+$ within the physiological range, and so we first examined the effect of changing extracellular pH ($pH_e$). The $pK_a$ of TASK-1 is 7.4, so to avoid any possible effect of changes in $pH_e$ on the ATP ligand itself ($pK_a$ 6.4), we compared GqPCR inhibition at $pH_e$ 8.5 compared to $pH_e$ 7.4. Interestingly, we found that the extent of GqPCR inhibition was reduced upon channel activation at $pH_e$ 8.5 (Fig. 2A).

Also, no small molecule drugs have currently been shown to directly activate TASK-1 channels via an increase in $P_o$. However, a class of negatively-charged activators (NCAs) directly activate many different $K_{2P}$ channels, but the most common, BL-1249, has little effect on TASK channels (Schewe et al, 2019). We therefore examined whether other ligands with a similar chemical structure to the NCA pharmacophore could directly activate TASK-1. We found that a related compound, NS3623 that has previously been shown to activate Kv4.3 and hERG $K^+$ channels (Lainez et al, 2018), potently activated TASK-1 in excised patches (Fig. 2B,C). Single-

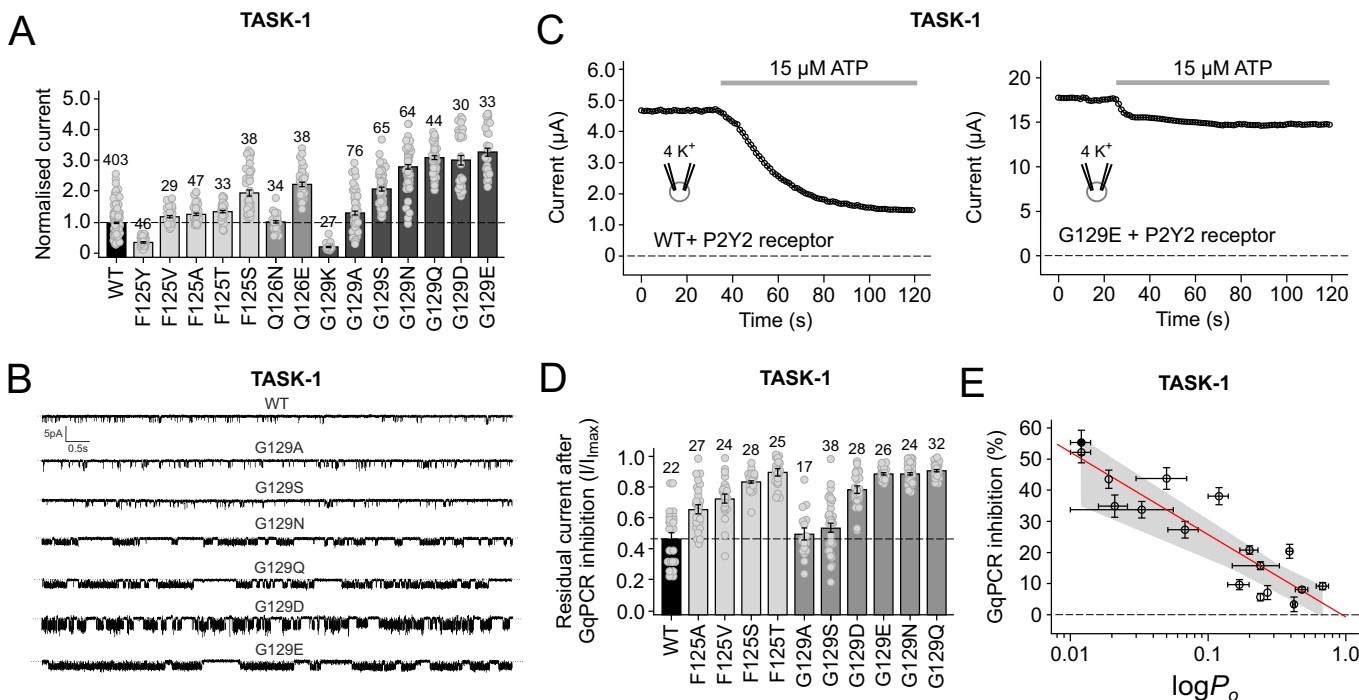

**Figure 1. Correlation between TASK-1 GqPCR sensitivity and channel $P_o$.**

(A) Macroscopic whole-cell currents for different mutations at positions F125, Q126 and G129. (B) Representative single-channel recordings of WT and mutant TASK-1 channels in cell-attached patches recorded at −100 mV showing a range of $P_o$. (C) Representative recordings showing GqPCR inhibition of WT and G129E mutant channels. TEVC whole-cell current levels are shown before and after (grey bar) extracellular application of 15 μM ATP to activate the co-expressed P2Y2 receptor. (D) Relative GqPCR inhibition for the GoF variants shown in (A). (E) Steep inverse correlation ($y = −26.49; R^2 = −0.79$) between GqPCR inhibition and single-channel $P_o$ for mutants in Table EV1. The linear regression is shown as red line and the 95% confidence intervals of the fit shown as the shaded grey area. WT TASK-1 is shown as a solid black circle. Data information: In (A, D), column bars display mean values with standard error (standard deviation/root($n$)) based on the indicated number of individual experiments ($n$; number of individual oocytes). The WT effect is highlighted as dashed lines. In (E) circles represent mean values with standard errors (data from Table EV1). Source data are available online for this figure.

channel recordings in cell-attached configuration at −100 mV also showed that this activation directly increased channel $P_o$ from $0.022 ± 0.003$ ($n = 10$) to $0.49 ± 0.06$ ($n = 10$).

We therefore measured whether activation by NS3623 also shifted GqPCR sensitivity. Consistent with our findings, we found that NS3623 markedly reduced the extent of GqPCR-mediated inhibition on whole-cell currents (Fig. 2D). Overall, these results are therefore consistent with a state-dependent mechanism for GqPCR-inhibition of TASK-1.

## GqPCR inhibition correlates with changes in sensitivity to inhibition by DAG

GqPCR-mediated inhibition of TASK channels is via direct interaction of DAG, and we have previously shown that the DDSA mutation, N133S, which has a reduced GqPCR sensitivity, exhibits a correspondingly dramatic reduction in sensitivity to inhibition by a short acyl chain derivative of DAG, i.e. 1,2-dioctanoyl-sn-glycerol (DiC8) when applied directly to excised patches from the intracellular side (Sormann et al, 2022). If the effect of this interaction is state-dependent, we would expect those mutations with a range of single-channel $P_o$ to exhibit corresponding reductions in their sensitivity to DAG.

We therefore examined DiC8 inhibition on different substitutions at G129, which exhibit a variety of $P_o$ and GPCR sensitivities. These mutations revealed a range of reduced sensitivity to DiC8 and $IC_{50}$ values that correlated with their relative increase in $P_o$ (Fig. 3A–C). We next investigated whether dynamic modulation of WT TASK-1 activity also affected DiC8 inhibition in excised patches and found that NS3623 activation markedly reduced this inhibitory effect (Fig. 3D,E).

Overall, these effects strongly support a mechanism for state-dependent inhibition of TASK channels via a direct interaction with DAG. However, such recordings of macroscopic currents cannot distinguish between the open and closed states of TASK channels. Therefore, to further evaluate this mechanism, we examined the inhibitory effect of DiC8 on TASK-1 at the single-channel level.

## DAG destabilises the open state of TASK-1

If DAG inhibits TASK channel activity via a reduction in $P_o$, then this can occur in at least two ways: either via stabilisation of the closed states of the channel, or by destabilising the open states because the resulting macroscopic effect will be the same. However, TASK-1 has a very low intrinsic channel $P_o$ and so to measure the effects of DAG inhibition on single WT channels would be

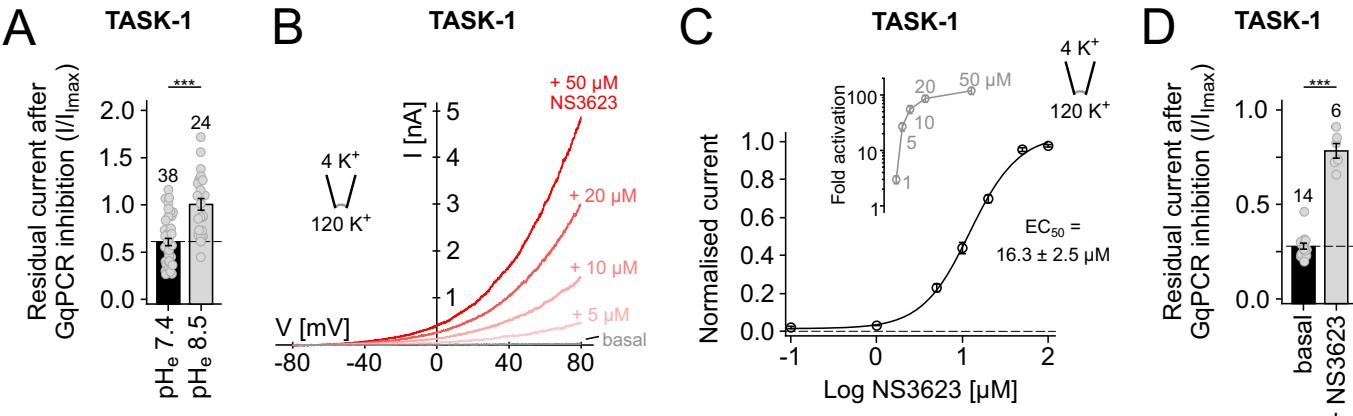

**Figure 2. Dynamic modulation of channel activity by $pH_e$ and NS3623 alters GqPCR-sensitivity.**

(A) Residual whole-cell current after GqPCR inhibition recorded at $pH_e$ 7.4 vs $pH_e$ 8.5, where TASK-1 activity is higher ($p < 0.001$), and GqPCR inhibition is reduced. (B) Activatory effect of different concentrations of NS3623 on WT TASK-1 currents recorded in giant excised patches. (C) Dose-response curve for TASK-1 activation by NS3623. Inset (grey) shows the fold-activatory effect of the indicated concentrations. (D) Whole-cell WT TASK-1 currents activated by NS3623 (50 μM) exhibit a markedly reduced GqPCR inhibition ($p < 0.001$). Data information: In (A, D), column bars display mean values with standard error (standard deviation/root(n)) based on the indicated number of individual experiments (n; oocytes). WT (basal, $pH_e$ 7.4) effect is highlighted as dashed lines. Significance of changes were determined using unpaired $t$-test. ***$p = 7.14*10^{-6}$ (A); ***$p = 6.80*10^{-6}$ (B). Source data are available online for this figure.

challenging. We therefore examined the effect of DiC8 on TASK-1 N133S mutant channels in excised patches because this mutant has a $P_o \sim 0.2$ compared to ~0.01 for WT TASK-1, but can still be completely inhibited by DiC8, albeit at much higher concentrations than for WT TASK-1 (Sormann et al, 2022). This showed that application of 1 μM DiC8 produced a dramatic reduction in single-channel $P_o$. (Fig. 4A)

Comparison of the dwell-time histograms for TASK-1 N133S before and after application of DiC8 reveals that it substantially reduced the duration of openings and burst durations (Fig. 4B and Table EV2). The distribution of closed times consists of two major peaks, which correspond to the intra-burst and inter-burst closed states, and a small population of very long closed states. DiC8 did not appear to noticeably alter the relative position of these two major peaks; however, it markedly reduced the number of short (intra-burst) closed states and concomitantly increased the number of long (inter-burst) closed states.

This suggests that open state destabilisation by DiC8 leads to more frequent exits from the open states to inter-burst closed states which dramatically increases their number (Fig. 4B). This is not the case for the short-closed states within bursts, thus their relative number in the presence of DiC8 are diminished. This effect also contributes to the overall reduction in burst durations (Fig. 4B). A more precise analysis of the intra-burst closures is limited by the temporal resolution of the acquired data; however, their potential destabilisation would further reduce both their relative number as well as the length of bursts. Overall, these results therefore imply that DiC8 inhibits TASK channels primarily via destabilisation of the open states and bursts of channel openings.

## DAG binds to TASK-1 in a groove between M2, M3 and M4

Our results support a mechanism where direct binding of DAG to TASK-1 reduces channel $P_o$, but the precise site of interaction

between DAG and TASK-1 remains unknown. High-resolution structural information now exists for both TASK-1 and TASK-3, and so we chose to take advantage of recent advances in the analysis of molecular dynamics (MD) simulation of lipid/protein interactions in membranes to probe for potential DAG binding sites.

PyLipID is a Python package that uses a community analysis approach for lipid binding site detection and calculating lipid residence times for both the individual protein residues and the detected binding sites (Song et al, 2022). Coarse-grained simulations were therefore initially performed with different numbers of DAG molecules in the lower leaflet. Potential interactions were then characterised by using PyLipID to identify residues with the highest residence time and occupancy. Representative binding poses were then converted into full-atom models for further atomistic simulations with DAG in the lower leaflet. A similar PyLipID analysis of these simulations revealed that the headgroup of DAG exhibits the highest residence time and occupancy with two main residues on M4, T230 and V234, both within a membrane-facing groove between the M2, M3 and M4 transmembrane helices of the same subunit of the channel (Fig. 5A).

This interaction site for DAG is positioned within the middle of the membrane, thus requiring movement of DAG from the inner leaflet towards this site after its generation by cleavage of $PIP_2$. However, consistent with the dynamic role this lipid plays in transmembrane signalling, a variety of experimental approaches have shown that DAG exhibits extremely rapid transbilayer movement in the millisecond timescale (Allan et al, 1978; Bennett and Tieleman, 2012; Campomanes et al, 2019; Hamilton et al, 1991). Consistent with this mobility, we also observed that varying the length and degree of saturation of the DAG acyl chains did not have any major influence on interaction with this site, though the acyl chains were capable of adopting several poses within this groove (Fig. EV1A).

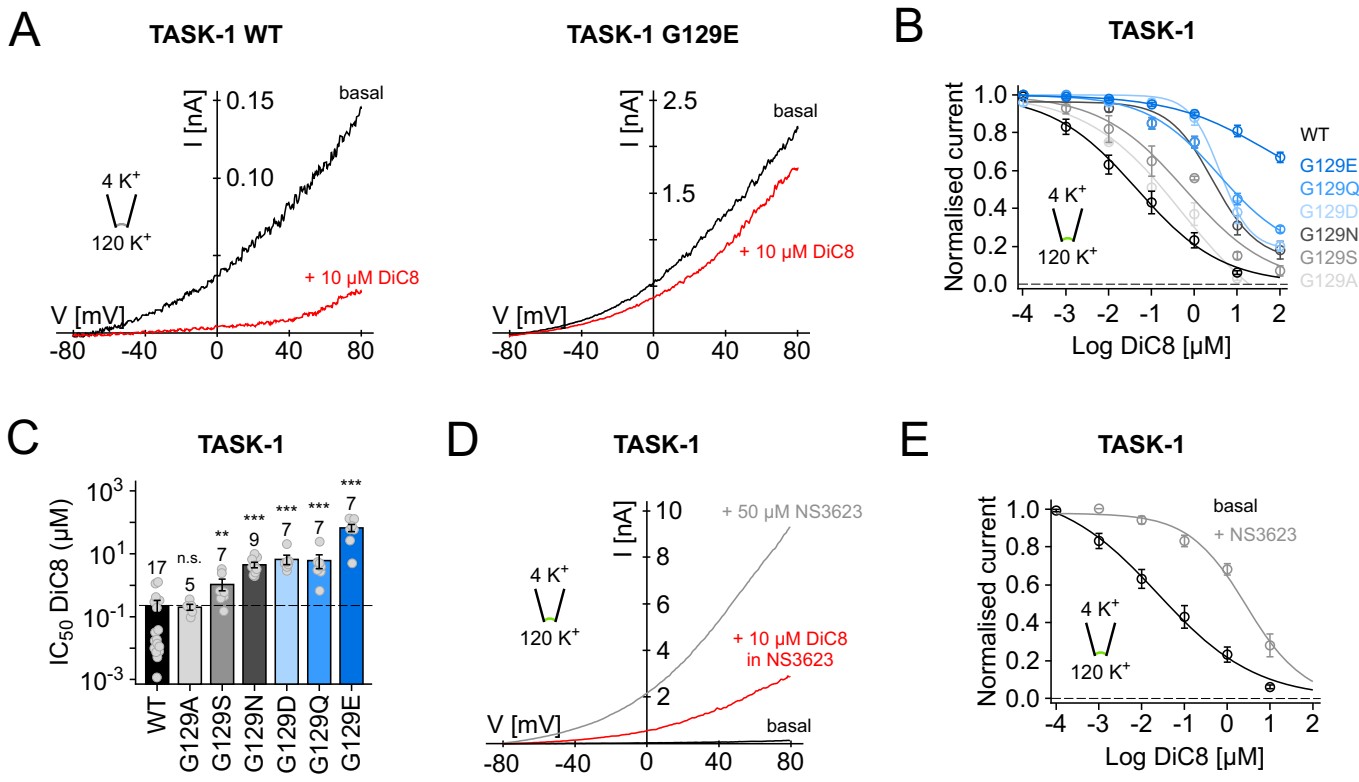

**Figure 3. Direct inhibition of WT and mutant TASK-1 channels by the synthetic DAG analogue DiC8.**

(A) Inhibition of WT TASK-1 and TASK-1 G129E by a short-chain DAG analogue (10 μM DiC8). Currents recorded in excised patches and DiC8 applied to the intracellular side of the respective patch. (B) Dose-response curves calculated for inhibition of WT and mutant TASK-1 channels, showing a variety of responses to DiC8. (C) IC$_{50}$ values for G129 mutants calculated from data shown in (B). (D) Activation of WT TASK-1 by NS3623 markedly reduces the inhibitory effect of 10 μM DiC8. (E) Dose-response curve showing the marked reduction in DiC8 inhibition after activation of WT TASK-1 with 50 μM NS3623. Data information: In (B, E) circles display mean values with standard error (standard deviation/root(n)) based on the number of individual experiments (n; excised patches). Lines represent Hill fits (method section). In (C), column bars display mean values with standard error based on the indicated number of individual experiments (n; excised patches). WT effect highlighted as a dashed line. Significance of changes determined using the Wilcoxon rank test. n.s. not significant (G129A); **$p = 0.003$ (G129S); ***$p = 3.02*10^{-6}$ (G129N); ***$p = 1.93*10^{-5}$ (G129D); ***$p = 4.48*10^{-5}$ (G129Q); ***$p = 1.93*10^{-5}$ (G129E). Source data are available online for this figure.

## Binding site analysis

To validate this interaction, we examined the effect of mutating key residues within this site, including both T230 and V234 in TASK-1 (Fig. 5A). However, our results in Fig. 1E show that any mutation which increases channel $P_o$, will reduce GqPCR inhibition irrespective of whether it may form part of the binding site or not. An unambiguous direct effect of a mutation on DAG binding can therefore only be inferred from a mutation which alters GqPCR sensitivity without significant changes in single-channel $P_o$. To help with this analysis, an initial screen was made for the effect of a range of mutations at key sites on M2 and M4 within this binding groove on both macroscopic whole-cell currents and GqPCR sensitivity before examining their single-channel $P_o$ (Fig. EV1B). From this initial screen, a number of mutations were identified which reduced GqPCR sensitivity consistent with their location within the binding site, but many also altered the whole-cell currents and/or were subsequently found to increase single channel $P_o$ thus complicating interpretation of their effect (Fig. EV1B).

However, both the V234L and T230V mutations on M4 within the proposed site (Fig. 5A) produced a marked reduction in whole-

cell GqPCR sensitivity with either a limited or no effect on single-channel $P_o$. The V234L mutation increased channel $P_o$ ~ 2-fold (Fig. EV2A), which would only have a modest effect on GqPCR sensitivity predicted by its change in $P_o$ (Fig. 1E). However, this mutation resulted in a >100-fold decrease in DAG sensitivity (Fig. EV2B). More significantly, no increase in $P_o$ at all was observed for the T230V mutation (Fig. 5C), yet it also resulted in a >100-fold decrease in DAG sensitivity (Fig. 5D,F). Importantly, both mutations retained their sensitivity to inhibition by external H$^+$, suggesting that their gating properties remain otherwise intact (Fig. EV2C). Together, the specific effect of mutations within this groove strongly supports a role for their interaction with DAG and its direct inhibitory effect.

## A DAG binding site that defines GqPCR sensitivity of both TASK-1 and TASK-3

Interestingly, TASK-1 and TASK-3 exhibit almost 70% amino acid sequence identity, especially within the core transmembrane regions. In particular, their sequence only differs by three amino acids within the M2 transmembrane helix, one of which is T230. In

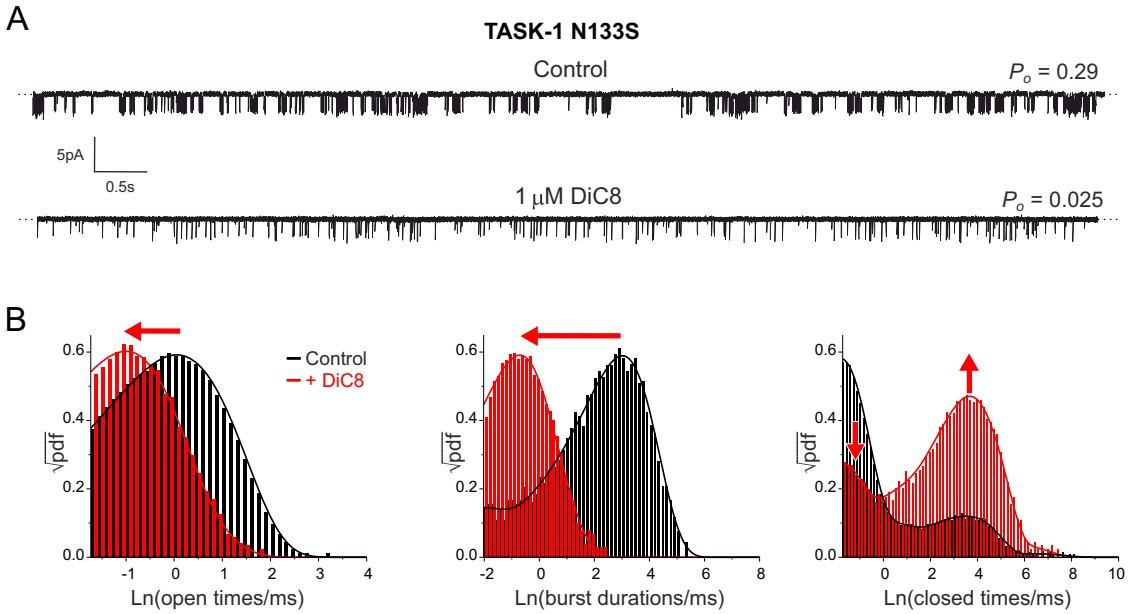

**Figure 4. DiC8 destabilises the open state of TASK-1.**

(**A**) Representative single-channel traces showing inhibition of N133S TASK-1 channels by 1 μM DiC8 applied to the inside of an excised patch (bottom trace). The sample traces are 1 s long, measured at −100 mV. (**B**) Dwell-time histograms (presented using Sine-Sigworth display) are obtained from the full recordings (>4 min.) either in the absence (black) or presence (red) of DiC8 and are overlayed for comparison. The apparent kinetic states of the channel manifest themselves as peaks in the histogram, with their mean lifetimes being the maximum peak values. The red arrows indicate main shifts and changes in the peak distributions resulting from DiC8 inhibition. Data information: In (**B**), three different types of distributions are shown as indicated: (Left) distribution of openings, (Middle) bursts of openings, and (Right) closings. The bursts are composite states consisting of openings and short closings contained within the peak on the far left in the distribution of closings. The lines are the best fit of probability density functions (methods section and Table EV2). Source data are available online for this figure.

TASK-3, this residue is a valine (V230), and WT TASK-3 channels have a reduced GqPCR sensitivity compared to TASK-1. We therefore examined the effect of the reverse (V230T) mutation in TASK-3 and found that it increased whole-cell GqPCR sensitivity with a concomitant 10-fold increase in DAG sensitivity (Fig. 5B,D,F). Importantly, the V230T mutation also did not affect whole-cell current levels or single-channel $P_o$ (Fig. 5B,E).

The overall effect of mutations within this site on DAG sensitivity clearly supports our identification of this site as a DAG binding site, but the ability of different mutations at a single site to either increase or decrease the direct inhibitory effect of DAG provide even stronger evidence. Furthermore, mutations at residues 230 also demonstrate that this site has the ability to fine-tune the relative sensitivity of TASK-1 vs. TASK-3 to GqPCR-mediated inhibition.

## Discussion

In this study, we have identified a site within the membrane-facing transmembrane domains of TASK K2P channels that is critical for GPCR-mediated regulation of channel activity. We show that this site interacts with DAG, a product of PIP₂ hydrolysis by PLC, to mediate the effect of Gαq-coupled receptors. Our results therefore provide a model whereby DAG interaction with this site results in a state-dependent inhibition, with DAG preferentially destabilising the open state of the channel to reduce channel $P_o$. We also show that this mechanism is the primary reason why all of the known

disease-causing GoF mutations produce channels with a markedly reduced GPCR sensitivity. This not only exacerbates the enhanced current levels produced by these mutations, but also, and perhaps more importantly, uncouples them from their regulation by native GqPCR-mediated signalling pathways. These results thus provide a greater insight into the mechanisms that control TASK K2P channel gating, and identify a common regulatory defect shared by this particular class of pathogenic GoF mutations.

Previous studies have shown that DAG directly inhibits both TASK-1 and TASK-3 channels (Sormann et al, 2022; Wilke et al, 2014), but the precise molecular mechanisms involved, and its site of action remained unclear. Likewise, it was not fully understood why all of the pathogenic GoF mutations, or indeed any mechanism that activates either TASK-1 or TASK-3 results in a dramatic reduction in GqPCR-mediated inhibition. Such GPCR regulation is critically important for neuronal activity in the control of sleep, chemosensation and ventilation, as well as for vasomotor tone and aldosterone secretion. The regulation of TASK channel activity is also being actively explored as a potential mechanism for the treatment of atrial fibrillation and sleep apnoea. Greater insight into these mechanisms of regulation, therefore, has the potential to improve the development of these channels as therapeutic targets.

The clear inverse correlation between single-channel $P_o$ and GPCR sensitivity is also reflected in corresponding reductions in DAG sensitivity because DAG inhibits via destabilisation of the open state. Consequently, dynamic modulation of TASK-1 channel activity by either $pH_e$ or by an agonist such as NS3623 also modulates GPCR inhibition via changes in DAG sensitivity due to this state-dependent mechanism of inhibition. This therefore

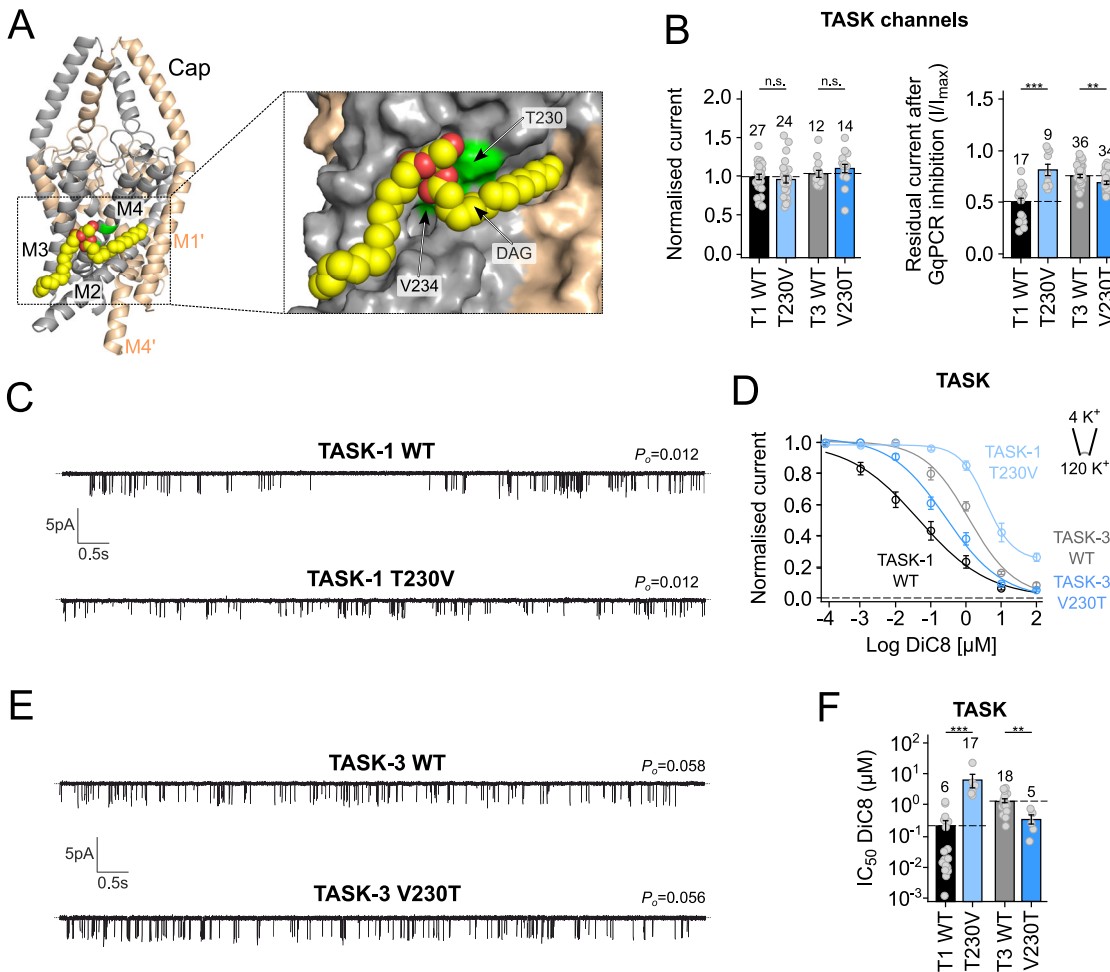

**Figure 5. A residue on M4 defines the GqPCR and DAG sensitivity of TASK channels.**

(**A**) Cartoon representation of the structure of TASK-1 with DAG bound in the site identified using PyLipID. The two subunits are coloured grey and wheat with DAG shown in yellow as vdW spheres with cpk colouring. DAG binds within a groove between M2, M3 and M4 of the same subunit. The expanded panel on the right shows the groove in more detail with TASK-1 in surface representation. The key residues on M4 which influence DAG inhibition (T230 and V234) are shown in green. (**B**) The T230V mutation in TASK-1 reduces GqPCR sensitivity but does not affect whole-cell currents. The reverse mutation (V230T) in TASK-3 does not affect whole-cell currents but increases the GqPCR sensitivity. (**C**) The TASK-1 T230V mutation does not affect single-channel $P_o$ (mean values of $P_o$ for WT and T230V mutant channels were $0.015 \pm 0.003$ ($n = 8$) and $0.012 \pm 0.003$ ($n = 8$), respectively). (**D**) Dose-response curves for WT and mutant TASK-1 and TASK-3 channels showing the effect of mutations on DiC8 sensitivity. (**E**) The V230T mutation in TASK-3 does not affect single channel $P_o$ (mean values of $P_o$ for WT and V230T mutant channels were $0.058 \pm 0.008$ ($n = 8$) and $0.056 \pm 0.012$ ($n = 8$), respectively). (**F**) $IC_{50}$ values analysed from measurements, as in panel (**D**), showing the opposing effect of mutations at residue 230 on DiC8 sensitivity in TASK-1 (T1) and TASK-3 (T3). Data information: In (**B**, **F**) column bars display mean values with standard error (standard deviation/root($n$)) based on the number of individual experiments ($n$; oocytes (**B**) or excised patches (**C**)). WT effects are highlighted as dashed lines. In (**D**), circles display mean values with standard error (standard deviation/root($n$)) based on individual experiments ($n$; number of excised patches recorded). Lines represent Hill fits (Method section). Significance of changes in (**B**) were determined using unpaired $t$-test. n. s. not significant; ***$p = 3.98*10^{-4}$ (T1); **$p = 0.006$ (T3). Significance of changes in (**F**) were determined using the Wilcoxon rank test. ***$p = 5.53*10^{-5}$ (T1); **$p = 0.004$ (T3). Source data are available online for this figure.

explains why channel activation, resulting from either pathogenic mutations or from dynamic channel modulation itself, reduces the apparent sensitivity of TASK channels to GPCR-mediated inhibition (Fig. 6).

In all of the pathogenic GoF mutations in TASK-1, there is a marked increase in channel $P_o$, and in the majority of cases, increased whole-cell currents as well. However, in some variants, e.g. L241F within the X-gate of TASK-1, there is little effect on the magnitude of whole-cell currents due to an associated reduction in channel trafficking to the plasma membrane (Rodstrom et al, 2020; Sormann et al, 2022). This therefore suggests that it may be the lack

of GqPCR regulation itself, rather than just the increased current levels, that contributes to their pathogenic effect.

The binding site for DAG that we identify is within an external membrane-facing groove formed by M2, M3 and M4 of the same subunit of TASK-1, including a critically important interaction with T230 on M4. Interestingly, threonine residues are not uncommon within transmembrane helices due to their ability to influence helical bends (Grey and Matthews, 1984). This site is also located just above the X-gate on M4 within the membrane and is therefore well placed to influence the relative stability of the open vs closed state of the channel.

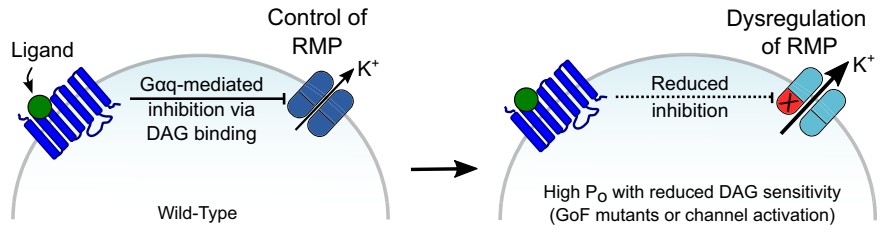

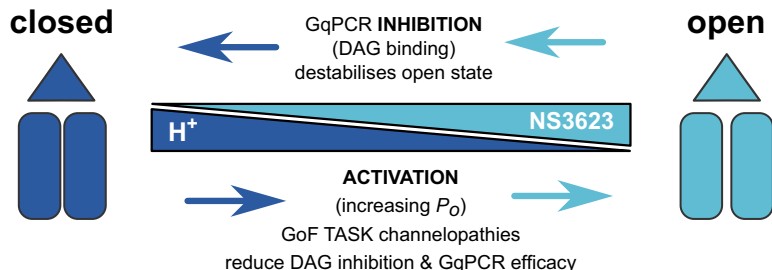

**Figure 6. A model for GqPCR regulation of TASK channels and their dysregulation in disease.**

(Upper panel) Cartoon describing GqPCR inhibition of TASK channels: GqPCR activation results in an increase in DAG levels (via hydrolysis of $PIP_2$) and direct inhibition of TASK activity. Inhibition of TASK $K^+$ channel activity results in depolarisation of the resting membrane potential (RMP), thus linking GqPCR activity to cellular electrical activity. DAG exhibits state-dependent regulation, so GoF mutations that increase $P_o$ reduce DAG inhibition, resulting in increased TASK channel activity and inability to control the RMP due to uncoupling from GqPCR pathways. (Lower panel) Dysregulation arises via state-dependent inhibition. DAG preferentially destabilises the open states of TASK channels, so any mutant that increases $P_o$ or a mechanism that increases activity (e.g. change in $pH_e$ or drugs such as NS3623) will reduce DAG-mediated inhibition and GqPCR efficacy. Data information: The images in each panel are schematics to illustrate and summarise the mechanisms involved. Source data are available online for this figure.

No structure yet exists for an open TASK channel, and so our analysis only examines the closed state, but the X-gate resides below T230 on M4, and only relatively small movements or rotations of M4 at this site are likely to be required to open the X-gate. Thus, the interaction of the DAG headgroup with T230, combined with other non-polar interactions, may therefore be sufficient to destabilise the open state. Importantly, T230 in TASK-1 is also one of the few amino acids that is different in TASK-3 within this region, and we show that exchanging this single residue has opposing effects on GqPCR sensitivity without any changes in $P_o$, thus explaining why TASK-3 channels are intrinsically less sensitive to GqPCR inhibition.

In conclusion, our findings indicate that DAG binds to an external-facing groove within the TM helices, to preferentially destabilise the open state and produce inhibition of TASK channels via GqPCR signalling. This mechanism not only explains the GPCR regulation of TASK channels, but also the common pathogenic effect of TASK channelopathies that result from channel dysregulation.

# Methods

**Reagents and tools table**

| Reagent/resource | Reference or source | Identifier or catalogue no. |
|---|---|---|
| **Experimental models** | | |
| *Xenopus laevis* oocytes | Xenopus 1 Corp. | 4230 |
| **Recombinant DNA** | | |

| Reagent/resource | Reference or source | Identifier or catalogue no. |
|---|---|---|
| Human TASK-1 in pFAW | Aryal et al, 2014 | NM_002246.3 |
| Human TASK-3 in pFAW | Aryal et al, 2014 | NM_001282534.2 |
| Human P2Y2 in pFAW | Sormann et al, 2022 | NM_002564.4 |
| **Antibodies** | | |
| N/A | | |
| **Oligonucleotides and other sequence-based reagents** | | |
| N/A | | |
| **Chemicals, enzymes and other reagents** | | |
| NS3623 | Tocris Bioscience | 4462 |
| 1,2-sn-dioctanoylglycerol (DiC8) | Merck Sigma-Aldrich | D0138 |
| **Software** | | |
| PyLipID | https://github.com/wlsong/PyLipID | N/A |
| Origin | https://www.originlab.com/ | Origin2021 |
| Clampfit | https://www.moleculardevices.com/ | pClamp11 |
| Clampex | https://www.moleculardevices.com/ | pClamp11 |
| Patchmaster | https://www.heka.com/index.html | 2x91 |

| Reagent/resource | Reference or source | Identifier or catalogue no. |
| --- | --- | --- |
| Igor | https://www.wavemetrics.com/ | Pro9 |
| **Other** | | |
| N/A | | |

## Molecular biology

The wild-type human TASK-1 gene (*KCNK3*) and human TASK-3 gene (*KCNK9*) in the pFAW dual-purpose oocyte expression vector was used throughout this study (Aryal et al, 2014). For GqPCR regulation, channels were co-expressed with the human P2Y2 receptor (Sormann et al, 2022). Mutations in TASK channels were introduced by site-directed mutagenesis and confirmed by sequencing. RNA was made using the T7 mScript Standard mRNA Production System for expression in *Xenopus* oocytes.

## Electrophysiology

### Whole-cell recordings

Oocytes from *Xenopus laevis* were used in this study. Mature female *Xenopus* were purchased from Xenopus 1 (USA) and kept in accordance with the UK Animals (Scientific Procedures) Act 1986 and local ethical approval (Biomedical Services Unit, University of Oxford, Oxford, UK). After removal via Schedule 1 procedure under terminal anaesthesia, each defolliculated oocyte was injected with 2 ng of mRNA and incubated for 20–24 h at 17.5 °C in Modified Barth's storage solution at pH 7.4 (88 mM NaCl, 1 mM KCl, 1.68 mM, $MgSO_4 7H_2O$, 10 mM HEPES and 2.4 mM $NaHCO_3$) supplemented with 0.05 mg/ml gentamicin. Recordings were performed in ND96 buffer at pH 7.4 (96 mM NaCl, 2 mM KCl, 2 mM $MgCl_2$, 1.8 mM $CaCl_2$, 5 mM HEPES). Whole-cell currents were recorded using a 400 ms voltage step protocol from a holding potential of −80 mV delivered in 10 mV increments between −120 mV and +50 mV. This was followed by an 800 ms ramp protocol from −120 to +50 mV. GqPCR inhibition studies were conducted as described before (Sormann et al, 2022) where currents at +50 mV were recorded before and 1 min. after perfusion with 15 μM ATP, a concentration that elicits 50% inhibition of WT TASK-1. All recorded traces were analysed using Clampfit (Axon Instruments), and graphs were plotted using Origin2021 (OriginLab Corporation). Whole-cell currents in response to $pH_e$ regulation was similarly recorded at +50 mV and compared before (at $pH_e$ 7.4) and 1 min. after perfusion with an extracellular recording solution adjusted to the stated pH. Due to the nature of the studies being performed, no data blinding was used in any electrophysiological recordings.

### Single-channel recordings

Single-channel currents were recorded with an Axopatch 200B amplifier via a Digidata 1440 A digitiser (Molecular Devices) at −100 mV, where the channels have sufficiently large single-channel conductance suitable for analysis of their gating. Data were filtered at 2 kHz and recorded at a 200 kHz sampling rate with Clampex (Molecular Devices). Pipette solution and bath solution for cell-attached experiments contained (in mM): 140 KCl, 2 $MgCl_2$, 1 $CaCl_2$, 10 HEPES (pH 7.4 adjusted with KOH/HCl). For inside-out experiments, the bath solution contained (in mM): 140 KCl, 2 $MgCl_2$, 1 $CaCl_2$, 10 HEPES (pH 7.2 adjusted with KOH/HCl). All experiments were conducted at room temperature. For analysis of single channels, open probability was calculated from idealised current traces constructed with a 50% threshold criterion with Clampfit (Molecular Devices) at an imposed resolution of 50 μs. Current data points below the threshold (half of the open level) were interpreted as closed state, and data points above the threshold as open state. Horizontal lines going through the means of the current values of individual openings and closings were generated using Clampfit. Summing the lengths of all openings and closings in the idealised recording allowed the programme to calculate $P_o$. Analysis of amplitude and dwell-time distributions was performed in Origin. The critical time for burst analysis was determined using Magleby and Pallotta criterion (Magleby and Pallotta, 1983). Statistical significance in DiC8 effects on channel lifetimes was determined by a paired *t*-test. Dwell-time histograms are presented using the Sine-Sigworth display, where the x-axis is the logarithm of a state duration and the y-axis is its corresponding square root number of occurrences in the recording. The apparent kinetic states of the channel manifest themselves as peaks in the histogram, with their mean lifetimes being the maximum peak values on the y-axis. Bursts are defined as composite states consisting of openings and short closings. The lines are the best fit of probability density functions $pdf(t) = \Sigma a_{i*} exp(-t/t_i)$ to the data where $t_i$ is a mean lifetime of state *i* and $a_i$ is the corresponding relative area of events of the state *i* in the distribution ($\Sigma a_i = 1$).

### Macroscopic patch recordings

Giant excised membrane patch measurements in inside-out configuration under voltage-clamp conditions were made at room temperature 72–120 h after injection of 50 nl channel specific mRNA into oocytes. Thick-walled borosilicate glass pipettes had resistances of 0.25–0.35 MΩ (tip diameter of ~15–30 μm) and were back-filled with extracellular solution containing (in mM): 4 KCl, 116 NMDG, 10 HEPES and 3.6 $CaCl_2$ (pH was adjusted to 7.4 with KOH/HCl). Bath solution was applied to the cytoplasmic side of the excised giant patches via a gravity flow multi-barrel application system and had the following composition (in mM): 120 KCl, 10 HEPES, 2 EGTA and 1 pyrophosphate. Currents were acquired with an EPC10 USB amplifier and HEKA PatchMaster 2 × 91. The sampling rate was 10 kHz, and the analogue filter was set to 3 kHz (−3 dB). Voltage ramp pulses (−80 mV to +80 mV) were applied from a holding potential ($V_H$) of −80 mV for 0.8 s with an inter-pulse interval of 9 s and were analyzed at a given voltage of +40 mV. The relative steady-state levels of inhibition for the indicated blocker were fitted with the following Hill equation:

$$Y = \frac{base + (max - base)}{\left(\frac{X_{half}}{X}\right)^H}$$

where base is the inhibited (zero) current, *max* is the maximum current, *X* is the blocker concentration, $X_{half}$ is the value of concentration for half-maximal occupancy of the blocker binding site and *H* is the Hill coefficient. 1,2-Dioctanoyl-*sn*-glycerol (DiC8) (Merck Sigma-Aldrich) and *N*-[4-Bromo-2-(1*H*-tetrazol-5-yl-phenyl]-*N'*-[3-(trifluoromethyl) phenyl]-urea (NS3623) (Tocris Bioscience) were stored as DMSO stock solutions (10–50 mM) at −80 °C and were diluted in the intracellular

bath solution to the final concentration prior to the measurements. To examine the direct effect of NS3623 activation on DiC8 inhibition, it was included in the internal recording solution for application to the intracellular side of excised patches.

## Molecular dynamics

In the coarse-grained simulations, three 20 μs simulation repeats were performed using the Martini 2.2 forcefield with an elastic network (EINeDyn) (de Jong et al, 2013). POPC membranes with different concentrations (5%, 10%, 20%, 33%) of DAG in the lower leaflet were tested. Binding sites were characterised by residues with the highest residence time and occupancy, employing a dual-distance cutoff approach with pyLipID, which uses a community analysis approach for binding site detection, calculating lipid residence times for the individual protein residues and the detected binding sites (Song et al, 2022). For representative binding poses generated with coarse-grained simulations, the cg2at.py tool was used to map coarse-grained beads to atoms and evaluate the stability of the binding poses with atomistic simulations (Vickery and Stansfeld, 2021). Atomistic simulations were then set up from scratch using Charmm GUI, with DAG concentrations (20%) in the lower leaflet and varying sidechain lengths and saturation (C12:0/C12:0, C14:0/C14:0, C16:0/16:0, C16:1/16:1 and C16:0/18:2). Simulations were performed with GROMACS 2021, using the TIP3P water model and the CHARMM36m forcefield (Huang et al, 2017). pyLipID was used to assess the DAG and protein interactions in three independent 5-μs-long atomistic repeats.

## Data availability

This study includes no data deposited in external repositories.

The source data of this paper are collected in the following database record: biostudies:S-SCDT-10_1038-S44318-026-00710-6.

## Peer review information

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

## Acknowledgements

This work was directly supported by grants from the Biotechnology and Biological Sciences Research Council and Medical Research Council to SJT and PCB (BB/T002018/1, BB/S008608/1, BB/S001247/1 and MR/W017741/1). It was also supported by the Wellcome Trust as part of the OXION Initiative in Ion Channels and Membrane Transport in Health and Disease (WT084655MA and 102161/B/13/Z). Further grants from the Deutsche Forschungsgemeinschaft supported the work of MS (SCHE 2112/1-2) and TB (BA 1793/6-2) as part of the Research Unit FOR2518, *Dynlon*. It was also supported by the Leibniz Collaborative Excellence Programme (K622/2024) to TB and MS. DS and KC are funded by the UKRI-BBSRC Interdisciplinary Bioscience Doctoral Training Partnership (BB/M011224/1).

## Author contributions

**Thibault R H Jouen-Tachoire**: Conceptualisation; Data curation; Formal analysis; Validation; Investigation; Methodology; Writing—original draft; Writing—review and editing. **Peter Proks**: Conceptualisation; Data curation; Formal analysis; Validation; Investigation; Visualisation; Methodology; Writing—review and editing. **David Seiferth**: Formal analysis; Validation; Investigation; Visualisation; Methodology; Writing—review and editing. **Kate Crowther**: Data curation; Formal analysis; Validation; Investigation; Visualisation; Methodology; Writing—review and editing. **Philip C Biggin**: Supervision; Funding acquisition; Writing—review and editing. **Thomas Baukrowitz**: Supervision; Funding acquisition; Validation; Writing—original draft; Writing—review and editing. **Marcus Schewe**: Conceptualisation; Data curation; Formal analysis; Supervision; Funding acquisition; Validation; Investigation; Visualisation; Methodology; Writing—original draft; Project administration; Writing—review and editing. **Stephen J Tucker**: Conceptualisation; Data curation; Formal analysis; Supervision; Funding acquisition; Validation; Investigation; Visualisation; Methodology; Writing—original draft; Project administration; Writing—review and editing.

Source data underlying figure panels in this paper may have individual authorship assigned. Where available, figure panel/source data authorship is listed in the following database record: biostudies:S-SCDT-10_1038-S44318-026-00710-6.

## Disclosure and competing interests statement

The authors declare no competing interests.

# Expanded View Figures

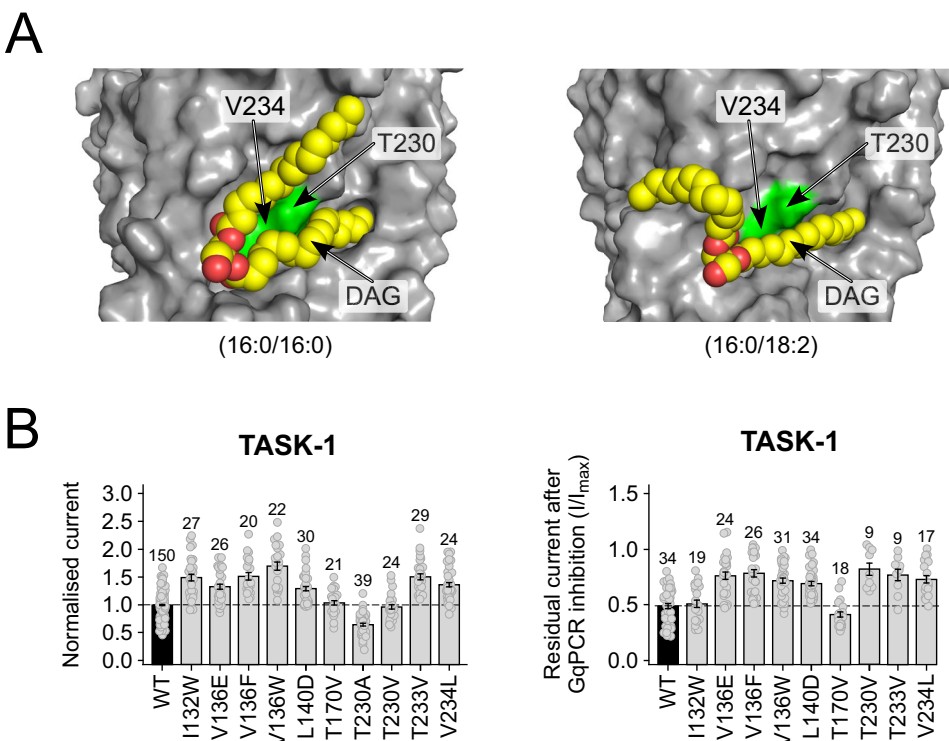

**Figure EV1. DAG sensitivity of TASK channels.**

(**A**) Less frequent poses of the acyl chains are also found for DAG in the groove between M2, M3 and M4 (see also Fig. 5A). Residues T230 and V234 are shown in green. (**B**) Normalised whole-cell current values and relative GqPCR sensitivity for mutations within this groove. Data information: In (**B**), column bars display mean values with standard error (standard deviation/root($n$)) based on the number of indicated experiments ($n$; number of individual oocytes). WT effects are highlighted as dashed lines. Source data are available online for this figure.

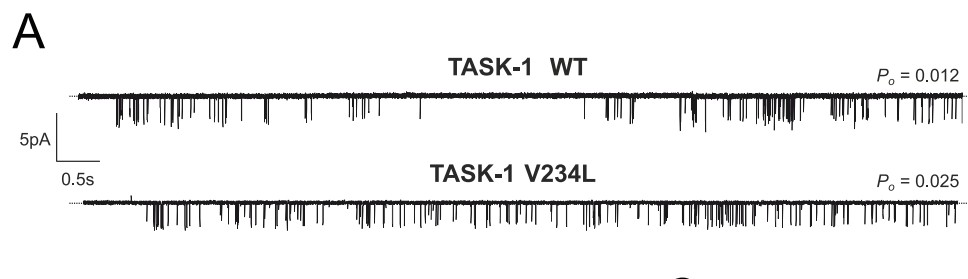

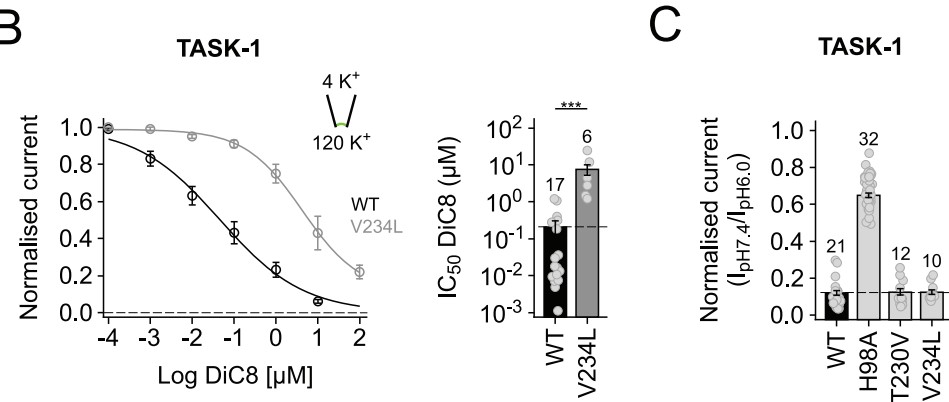

**Figure EV2.  Molecular determinants of DAG interaction with TASK channels.**

(A) Single-channel recordings for WT and V234L mutant TASK-1 channels. (B) This mutation only increases $P_o$ around twofold yet decreases DiC8 inhibition >100-fold as measured in excised patches. Left: Dose-response curve showing the marked reduction in DiC8 inhibition for V234L compared to WT TASK-1. Right: Reduced IC$_{50}$ values for DiC8 inhibition of V234L (***$p < 0.001$). (C) T230V and V234L mutant channels retain their sensitivity to inhibition by pH$_e$. The reduced sensitivity of the H98A pH sensor mutation is shown as a control. Data information: In (B), circles display mean values with standard error (standard deviation/root($n$)) based on the number of individual experiments ($n$; excised patches). Column bars display mean values with standard error (standard deviation/root($n$)) based on the number of indicated experiments ($n$; number of excised patches). Significance of changes were determined using the Wilcoxon rank test. ***$p = 5.53 \times 10^{-5}$. In (C), column bars display mean values with standard error (standard deviation/root($n$)) based on the number of indicated experiments ($n$; number of individual oocytes). In both, the WT effect is highlighted as a dashed line. Source data are available online for this figure.

