## [Peer Review File · The EMBO Journal]

Structural determinants for GPCR-mediated inhibition of TASK K2P channels by diacylglycerol and its dysfunction in disease

Thibault Jouen-Tachoire, Peter Proks, David Seiferth, Kate Crowther, Philip Biggin, Thomas Baukrowitz, Marcus Schewe, and Stephen Tucker

Corresponding author(s): Stephen Tucker (stephen.tucker@physics.ox.ac.uk) , Marcus Schewe (m.schewe@physiologie.uni-kiel.de)

Review Timeline:

Submission Date:	7th Aug 25
Editorial Decision:	1st Oct 25
Revision Received:	13th Dec 25
Editorial Decision:	14th Jan 26
Revision Received:	20th Jan 26
Accepted:	26th Jan 26

Editor: William Teale

Transaction Report:

Dear Stephen,

Thank you again for the submission of your manuscript entitled "Structural determinants for GqPCR-inhibition of TASK K2P channels and their dysfunction in disease" (EMBOJ-2025-112107) and for your patience during the review process. We have now received reports from two referees, which I copy below.

As you can see from their comments, both thought that your investigation was well-framed and timely. That said, both of them ask you to expand on your model for the DAG-TASK1 binding; this should be done before your manuscript can be published in The EMBO Journal. I think it would be useful if we were to discuss this once you have had an opportunity to digest the reports.

Based on the overall interest expressed in these reports, however, I would like to invite you to address the comments of all referees in a revised version of the manuscript. I should add that it is The EMBO Journal policy to allow only a single major round of revision and that it is therefore important to resolve the main concerns at this stage. I believe the concerns of the referees are reasonable and addressable, but please contact me if you have any questions, need further input on the referee comments or if you anticipate any problems in addressing any of their points. I am always available for a Zoom call. Please, follow the instructions below when preparing your manuscript for resubmission.

I would also like to point out that as a matter of policy, competing manuscripts published during this period will not be taken into consideration in our assessment of the novelty presented by your study ("scooping" protection). We have extended this 'scooping protection policy' beyond the usual 3 month revision timeline to cover the period required for a full revision to address the essential experimental issues. Please contact me if you see a paper with related content published elsewhere to discuss the appropriate course of action.

Again, please contact me at any time during revision if you need any help or have further questions.

Thank you very much again for the opportunity to consider your work for publication. I look forward to your revision.

Best regards,

William

William Teale, Ph.D.
Editor
The EMBO Journal

When submitting your revised manuscript, please carefully review the instructions below and include the following items:

- 1) a .docx formatted version of the manuscript text (including legends for main figures, EV figures and tables). Please make sure that the changes are highlighted to be clearly visible.
- 2) individual production quality figure files as .eps, .tif, .jpg (one file per figure).
- 3) a .docx formatted letter INCLUDING the reviewers' reports and your detailed point-by-point response to their comments. As part of the EMBO Press transparent editorial process, the point-by-point response is part of the Review Process File (RPF), which will be published alongside your paper.
- 4) a complete author checklist, which you can download from our author guidelines ([https://wol-prod-cdn.literatumonline.com/pb-assets/embo-site/Author Checklist%20-%20EMBO%20J-1561436015657.xlsx](https://wol-prod-cdn.literatumonline.com/pb-assets/embo-site/Author%20Checklist%20-%20EMBO%20J-1561436015657.xlsx)). Please insert information in the checklist that is also reflected in the manuscript. The completed author checklist will also be part of the RPF.
- 5) Please note that all corresponding authors are required to supply an ORCID ID for their name upon submission of a revised manuscript.
- 6) We require a 'Data Availability' section after the Materials and Methods. Before submitting your revision, primary datasets produced in this study need to be deposited in an appropriate public database, and the accession numbers and database listed under 'Data Availability'. Please remember to provide a reviewer password if the datasets are not yet public (see

<https://www.embopress.org/page/journal/14602075/authorguide#datadeposition>). If no data deposition in external databases is needed for this paper, please then state in this section: This study includes no data deposited in external repositories. Note that the Data Availability Section is restricted to new primary data that are part of this study.

Note - All links should resolve to a page where the data can be accessed.

8) For data quantification: please specify the name of the statistical test used to generate error bars and P values, the number (n) of independent experiments (specify technical or biological replicates) underlying each data point and the test used to calculate p-values in each figure legend. The figure legends should contain a basic description of n, P and the test applied. Graphs must include a description of the bars and the error bars (s.d., s.e.m.).

9) We would also encourage you to include the source data for figure panels that show essential data. Numerical data can be provided as individual .xls or .csv files (including a tab describing the data). For 'blots' or microscopy, uncropped images should be submitted (using a zip archive or a single pdf per main figure if multiple images need to be supplied for one panel). Additional information on source data and instruction on how to label the files are available at .

10) We replaced Supplementary Information with Expanded View (EV) Figures and Tables that are collapsible/expandable online (see examples in <https://www.embopress.org/doi/10.15252/embj.201695874>). A maximum of 5 EV Figures can be typeset. EV Figures should be cited as 'Figure EV1, Figure EV2" etc. in the text and their respective legends should be included in the main text after the legends of regular figures.

12) Our journal encourages inclusion of *data citations in the reference list* to directly cite datasets that were re-used and obtained from public databases. Data citations in the article text are distinct from normal bibliographical citations and should directly link to the database records from which the data can be accessed. In the main text, data citations are formatted as follows: "Data ref: Smith et al, 2001" or "Data ref: NCBI Sequence Read Archive PRJNA342805, 2017". In the Reference list, data citations must be labeled with "[DATASET]". A data reference must provide the database name, accession number/identifiers and a resolvable link to the landing page from which the data can be accessed at the end of the reference. Further instructions are available at .

13) In order to increase the reproducibility and reach of your work, The EMBO Journal includes a table of reagents that were used in the study. Please provide this along with your

Further instructions for preparing your revised manuscript:

We realize that it is difficult to revise to a specific deadline. In the interest of protecting the conceptual advance provided by the work, we recommend a revision within 3 months (30th Dec 2025). Please discuss the revision progress ahead of this time with the editor if you require more time to complete the revisions. Use the link below to submit your revision:

Referee #1:

Jouen-Tachoire and colleagues follow up on their earlier work in which they identified mutations in the TASK1 channel that increase channel activity, and are associated with a developmental disorder with sleep apnea in humans. These mutant channels are not only overactive, but they also lack receptor-mediated inhibition. In the current work, they introduce additional mutations in the three residues where mutations were found in patients, with the goal of creating a spectrum of channels with different activities, including ones that are intermediate between the activity level of wild-type and disease associated overactive channels. One of the key findings of the manuscript is that there is a strong negative correlation between single channel open probability and GqPCR inhibition, indicating that the lack of GPCR inhibition is likely the consequence of increased stability of the open state. Additional experiments further solidify this conclusion, including negative correlation of DAG (DiC8) inhibition and open probability and reduced GPCR inhibition when the channel is stimulated by NS3623. Additionally, they identify a potential binding site for DAG using computational approaches, and identify a mutant in this binding site, that shows reduced inhibition by DAG, but its open probability is the same as the wild type channel. Overall, in this reviewer's opinion, the conclusion of the first part is solid, but the result was quite predictable, if you increase the open state stability of a channel, either by mutations, or by activators, inhibition usually decreases. The putative DAG binding site data are more novel, but this part is less developed. Specific comments are below:

1. The conclusion that reduced GPCR and DAG inhibition is due to increase open state stability is solid, but not very surprising. In this reviewer's opinion it provides incremental conceptual advance.
2. The DAG binding site data is more conceptually novel, but it is less developed. For example, it is based on a single mutant (T230V) which shows no increase in open probability, but shows reduced DAG inhibition. These data are compatible with this residue being part of the DAG binding site, but it is also open for alternative interpretations. How did the authors pick which residue to mutate T230 to? Was DAG binding reduced in silico?
3. The authors mention that several mutations were found that reduced GPCR inhibition, but they also increased open probability. Were they also in the putative DAG binding site?
4. Data presentation needs to be improved, as in the current version, it is hard to tell how some of the experiments were performed, which would be a hindrance if anyone tried to replicate the data. For example, there are no representative traces for GqPCR inhibition. Even though a complicated voltage protocol is described in the methods, it is unclear what voltage the currents were measured for data analysis, how long ATP was applied etc.
5. Figure 2: The effect of NS3623 was measured in macropatches, but unclear how it was applied for GPCR inhibition.
6. There are also no statistical analysis and significance calculations in any of the figures.

7. The title needs to be changed, as in its current version it is misleading, and sounds as if there was structural work involved.

Minor comments:

8. The scatter plots make the figures very crowded, and in many cases it impossible to tell what the mean of the data is. The authors should find a way to remediate this, for example they could decreased the symbol size of the scatters, an/or place the scatter plot next to the bar - I think the Origin Software offers this option.

9. Provide proper citation for Crowther 2025

10. I assume the n-s refer to the number of oocytes. Please also disclose the number of independent oocyte preparations

Referee #2:

Jouen-Tachoire and colleagues report a mutational and computational study aimed at defining the role of diacylglycerol (DAG), a product of PIP2 hydrolysis by GPCR activation of phospholipase C, in affecting the function of the K2P channel TASK1. The authors provide functional analysis (both whole cell and single channel) that supports the idea that DAG stabilizes the closed state of the channel. Computational studies identify a groove between the M2, M3, and M4 helices, located roughly mid-membrane with the hydrophilic part of DAG oriented towards the extracellular side. The authors show that sequence variation between TASK1 and TASK3 in this site contributes to the differential sensitivity of TASK1 (sensitive, Thr) and TASK3 (reduced sensitivity, Val). Overall, the combined studies provide good evidence for this unusual site.

Have only two comments to help clarify the work. The first one is particularly important as there are key mechanistic implications to the non-straightforward modulation mechanism proposed in this study.

1) The orientation of the DAG site requires that the DAG (liberated on the intracellular side of the membrane) needs to do some molecular gymnastics to reach its site. This necessarily involves moving the hydrophilic glycerol headgroup of the lipid from the intracellular side of the membrane to an orientation that faces the extracellular side. This key point is not clearly addressed. The authors do note in the discussion that a computational study (Campomanes et al 2019) suggests that DAG can accumulate between the bilayer leaflets. While this reference provides some support, most readers are going to miss this key speculative point of the current manuscript as the challenge to flipping DAG into the middle of the bilayer is never directly mentioned. The authors need to be more explicit in pointing out what needs to happen for DAG to reach its proposed site. It would also be good if they could provide other references. If there are any experimental ones, in particular that support DAG flipping in the membrane, they should be referenced, otherwise the evidence is a computational study here and a prior computational study.

I think that the proposal is novel and certainly fits the available data and models. The coincidence of identifying a residue in the TASK1/3 sensitivity differences is good support; however, the ease with which DAG can do the necessary gymnastics (and put a hydrophilic moiety in the middle of the bilayer) is going to have limiting effects on the efficacy of the proposed mechanism. This limitation needs to be directly addressed.

2) I missed the information on which tests were used to calculate the p values in the figures.

EMBOJ-2025-122107 Response to Reviewers Comments

Referee #1:

Jouen-Tachoire and colleagues follow up on their earlier work in which they identified mutations in the TASK1 channel that increase channel activity, and are associated with a developmental disorder with sleep apnea in humans. These mutant channels are not only overactive, but they also lack receptor-mediated inhibition. In the current work, they introduce additional mutations in the three residues where mutations were found in patients, with the goal of creating a spectrum of channels with different activities, including ones that are intermediate between the activity level of wild-type and disease associated overactive channels. One of the key findings of the manuscript is that there is a strong negative correlation between single channel open probability and GqPCR inhibition, indicating that the lack of GPCR inhibition is likely the consequence of increased stability of the open state. Additional experiments further solidify this conclusion, including negative correlation of DAG (DiC8) inhibition and open probability and reduced GPCR inhibition when the channel is stimulated by NS3623. Additionally, they identify a potential binding site for DAG using computational approaches, and identify a mutant in this binding site, that shows reduced inhibition by DAG, but its open probability is the same as the wild type channel. Overall, in this reviewer's opinion, the conclusion of the first part is solid, but the result was quite predictable, if you increase the open state stability of a channel, either by mutations, or by activators, inhibition usually decreases. The putative DAG binding site data are more novel, but this part is less developed. Specific comments are below:

1. The conclusion that reduced GPCR and DAG inhibition is due to increase open state stability is solid, but not very surprising. In this reviewer's opinion it provides incremental conceptual advance.

We thank the reviewer for their positive comments about our work. However, whilst we agree that state-dependence of ligand action has been observed before in some other ion channels, and that reduced agonist efficacy often correlates with increased P_o , we do not consider our specific results easily predictable. We and others have previously shown that Norfluoxetine block of filter gating in another K2P channel is state-independent (e.g. Proks, *JGP*, 2021) and there are other examples of state-independent blockers, especially those involved in pore block (e.g. the many studies of QA block of MthK) as well as for bipuvacaine block of NMDA receptors (Paganelli and Popescu, *J Neuroscience* 2015), so such results are not necessarily easy to predict. Furthermore, the precise mechanism of DAG action has not been directly addressed and the inhibition need not necessarily have exhibited any state-dependence at all, let alone such a clear effect. More importantly, our analysis of this state-dependence represents the first, but necessary, part of this study, and without demonstrating this clear effect it would not have been possible to draw the important conclusions we are then able to make. A full characterisation of this state-dependent effect therefore represents an appropriate and critical part of this study.

Overall, our results explain a number of previously unknown features, as well as the common effect of a diverse range of highly pathogenic Gain-of-Function variants on GPCR-mediated inhibition – a molecular defect that has also not been properly addressed before. Importantly, this has also allowed us to identify the structural features involved in the inhibitory effect of DAG.

2. The DAG binding site data is more conceptually novel, but it is less developed. For example, it is based on a single mutant (T230V) which shows no increase in open probability, but shows reduced DAG inhibition. These data are compatible with this residue being part of the DAG binding site, but it is also open for alternative interpretations. How did the authors pick which residue to mutate T230 to? Was DAG binding reduced in silico?

We are pleased that the reviewer finds this part of the study interesting. But importantly, our validation of this binding site is not based upon a single piece of evidence or even a single mutation, but built using

increasing layers of supporting data that concludes with a computational analysis of lipid interactions (PyLipID analysis of DAG interactions with TASK1) and functional validation of the most likely binding site predicted by these interactions.

As with any newly proposed binding site, even seemingly obvious ones revealed by crystallography or cryoEM, some form of functional validation is always necessary and we focussed on a key mutation at T230. However, we apologise if the reviewer was under the misapprehension that only the T230V mutation was used to support our hypothesis. We have therefore made this much clearer in our revised manuscript with a new separate subheading covering the mutagenic validation.

Based upon our criteria that a mutation in the binding site should not significantly affect P_o if we are to draw major conclusions, TASK1 T230V was the primary mutation analysed because it is a significant chemical change and the equivalent residue in TASK3 is a valine so this represents one of the key differences between TASK-1 and TASK3 within this region.

Importantly, other mutations were indeed tested at this site, for example, T230A is shown in ExtFig.EV1b (previously S1b) which resulted in reduced whole cell currents thus making further analysis of inhibition of these small currents by GqPCR inhibition impractical and so was not pursued. However, The T230V mutation not only showed normal single channel characteristics, but also a markedly reduced GqPCR and DAG-sensitivity demonstrating that the effects were unlikely due to non-specific allosteric changes and a change in gating kinetics

Another critical mutation within this groove (V234L) was also examined and we apologise if its importance was not made clear. This is also described in Fig.5 and ExtFig. EV1. This mutation also markedly reduced GqPCR/DAG-sensitivity and is located very close to T230. However, it results in a very small increase in P_o (approx. 2-fold) and so does not perfectly address our initial strict selection criteria. Nevertheless, this relatively small increase in P_o cannot account for the 50 to 100-fold decrease in DAG sensitivity the mutant produces, and so is likely to result from a change in the binding site as well. This point was only discussed briefly in the previous version of the ms and so we have now expanded and clarified this in the new section covering validation of the binding site. Several other mutations were also examined which are also consistent with our conclusions (see ExtFig1.EV1 and Comment 3 below for details).

However, of most interest was that the reverse mutation in TASK-3 (V230T) switched the GqPCR/DAG-sensitivity in the opposite direction. Normally, mutations within a predicted binding site that reduce ligand sensitivity are taken as important validation of that site, but the identification of a specific residue (i.e., T230) within a predicted binding site where ligand sensitivity can be switched in the **opposite** direction by mutagenesis would normally be taken as strong validation of the importance of that site. Of course, alternative and more complex interpretations can always be created for any result, and we are well aware of the age-old problem of interpreting changes in affinity vs efficacy, but in this case our conclusions represent the simplest interpretation of the results.

3. The authors mention that several mutations were found that reduced GPCR inhibition, but they also increased open probability. Were they also in the putative DAG binding site?

One of the major questions we wished to address was why a large range of highly pathogenic GoF mutations located in different regions of the channel all reduce GPCR inhibition – and we show that this is due to the state-dependence of inhibition via their large increase in P_o . We therefore had to be very careful in interpreting the effect of mutations within our predicted DAG binding site. As part of our screen that focussed on residues within the predicted site, several other mutations were indeed identified that affected GPCR inhibition. In addition to the V234L mutation discussed above, several other residues within this DAG binding site were also examined including F125, I132 and V136 on M2, and T233 on M4 – these mutations are shown in ExtFig.EV1 and Table EV1. All of these mutations resulted in decreased GqPCR inhibition (and/or direct DAG inhibition), but also exhibited a marked increase in P_o – so did not

meet our strict criteria for interpretation. Such effects are also not unexpected for mutations within a gating critical region and so are still consistent with their presence within the binding site, even if they do not provide unequivocal direct support. We just wanted to be careful that we did not fall into the common trap of overstating that their effects directly validate the proposed binding site, hence our original focus on T230 and V234.

4. Data presentation needs to be improved, as in the current version, it is hard to tell how some of the experiments were performed, which would be a hinderance if anyone tried to replicate the data. For example, there are no representative traces for GqPCR inhibition. Even though a complicated voltage protocol is described in the methods, it is unclear what voltage the currents were measured for data analysis, how long ATP was applied etc.

We apologise for the apparent lack of detail some of which was buried in the methods - many of these experimental approaches are also identical to those we recently reported (Sörmann et al, *Nature Genetics* 2021 and elsewhere) and so were not described in detail. However, representative traces for GqPCR mediated inhibition are now included in a new panel Fig.1b, and a more detailed experimental description about GqPCR inhibition via the coexpressed P2Y2 receptor is now included within the methods section and the figures themselves.

5. Figure 2: The effect of NS3623 was measured in macropatches, but unclear how it was applied for GPCR inhibition.

The small molecule agonist, NS3623 was applied directly by perfusion of the intracellular surface of the patch in excised patches, whilst for whole cell GqPCR inhibition it was applied by direct perfusion of the extracellular recording solution in two-electrode voltage clamp. This has now been clarified within the figure legends and the methods section and the type of experiment (whole-cell vs excised patch) is indicated more clearly by descriptive icons in the relevant figure panels.

6. There are also no statistical analysis and significance calculations in any of the figures.

This has now been corrected to make the statistical significance of the key results, *p*-values and tests used etc more visible in the figures, legends, methods etc. in accordance with the formatting guidelines. All data points are also included in the source data file along with reports of the *n* numbers and SEM data values. We have also improved the figures so the individual data points shown do not obscure the errors bars which are now more visible (but in most cases also still very small).

7. The title needs to be changed, as in its current version it is misleading, and sounds as if there was structural work involved.

We respectfully disagree on this point. We were careful not to use any phrase within the title that could suggest that we used more traditional (i.e., non-computational) 'structural' approaches to determine the residues and regions of the structure involved. Computational prediction of structures and binding sites combined with mutagenic validation represents a valid scientific approach, and so we feel that the use of the term 'structural determinants' in the title remains both valid and informative without being misleading. Furthermore, the abstract clearly describes the experimental scope of the study and is therefore unambiguous.

Minor comments:

8. The scatter plots make the figures very crowded, and in many cases it impossible to tell what the mean of the data is. The authors should find a way to remediate this, for example they could decreased the symbol size of the scatters, an/or place the scatter plot next to the bar - I think the Origin Software offers this option.

We have replotted the relevant figures to align the data points and also made the errors bars more visible (please note that in most cases they are very small). More statistical information is also provided in the figure legends according to the standard formatting guidelines and the mean values are also quoted in some of the accompanying tables, as well as in the source data files.

9. Provide proper citation for Crowther 2025

This reference has now been updated.

10. I assume the n-s refer to the number of oocytes. Please also disclose the number of independent oocyte preparations

In all cases the number of independent preparations was ≥ 3 . This is now clearly stated in the methods section and all data are now shown in the source data files.

Referee #2:

Jouen-Tachoire and colleagues report a mutational and computational study aimed at defining the role of diacylglycerol (DAG), a product of PIP2 hydrolysis by GPCR activation of phospholipase C, in affecting the function of the K2P channel TASK1. The authors provide functional analysis (both whole cell and single channel) that supports the idea that DAG stabilizes the closed state of the channel. Computational studies identify a groove between the M2, M3, and M4 helices, located roughly mid-membrane with the hydrophilic part of DAG oriented towards the extracellular side. The authors show that sequence variation between TASK1 and TASK3 in this site contributes to the differential sensitivity of TASK1 (sensitive, Thr) and TASK3 (reduced sensitivity, Val). Overall, the combined studies provide good evidence for this unusual site. Have only two comments to help clarify the work. The first one is particularly important as there are key mechanistic implications to the non-straightforward modulation mechanism proposed in this study.

We thank the referee for their extremely positive assessment of the studies included in the manuscript.

1) The orientation of the DAG site requires that the DAG (liberated on the intracellular side of the membrane) needs to do some molecular gymnastics to reach its site. This necessarily involves moving the hydrophilic glycerol headgroup of the lipid from the intracellular side of the membrane to an orientation that faces the extracellular side. This key point is not clearly addressed. The authors do note in the discussion that a computational study (Campomanes et al 2019) suggests that DAG can accumulate between the bilayer leaflets. While this reference provides some support, most readers are going to miss this key speculative point of the current manuscript as the challenge to flipping DAG into the middle of the bilayer is never directly mentioned. The authors need to be more explicit in pointing out what needs to happen for DAG to reach its proposed site. It would also be good if they could provide other references. If there are any experimental ones, in particular that support DAG flipping in the membrane, they should be referenced, otherwise the evidence is a computational study here and a prior computational study.

I think that the proposal is novel and certainly fits the available data and models. The coincidence of identifying a residue in the TASK1/3 sensitivity differences is good support; however, the ease with which DAG can do the necessary gymnastics (and put a hydrophilic moiety in the middle of the bilayer) is going to have limiting effects on the efficacy of the proposed mechanism. This limitation needs to be directly addressed.

For the sake of brevity, our original discussion only briefly addressed the issue of DAG movement within bilayer and only a single, more recent reference was quoted. We therefore thank the reviewer for the opportunity to expand on this point.

DAG is obviously an important signalling lipid and its transbilayer movement has been extensively studied over the last 50 years as this underlies its ability to rapidly mediate a wide variety of cellular signalling process. A range of experimental approaches have previously been used to demonstrate this including chemical modification & trapping (Alan et al, *Nature* 1978), ¹³C NMR spectroscopy (Hamilton et al, *JBC* 1991), and Electron Spin resonance approaches using labelled lipid analogs (Pohl, *Mol Memb Biol* 2009), as well as other computational approaches (Bennett, *J Lipid Res*, 2012). The general conclusion of these studies is that transbilayer diffusion of these lipids from one leaflet to another is extremely rapid with upper estimates in the range of 10-50 milliseconds for normal membrane bilayers (and even as fast as 30 microseconds in a pure POPC bilayer). The single reference cited (Campomanes et al 2019) was originally chosen because this demonstrates that DAG can also accumulate between leaflets which is precisely where the binding site for DAG is predicted on TASK1. We have therefore expanded this section of the discussion and included additional references to more direct experimental approaches which demonstrates this rapid diffusion.

2) I missed the information on which tests were used to calculate the p values in the figures.

The tests used are now stated more clearly within the methods sections and the statistical significance of the key results are shown more prominently in the relevant figure legends and source data files.

Dear Stephen,

We have now received re-review reports from both referees, which I have included below. As you will see, you have addressed their concerns satisfactorily. Before I can finally accept the manuscript, there are some remaining editorial points which need to be addressed. In this regard would you please:

- remove EV figures from the main manuscript text and provide their legends after the references,
 - remove the AC/CrediT section from the text,
 - acknowledge funding from the following in our online submission system: BB/T002018/1, BB/S008608/1, BB/S001247/1 and MR/W017741/1; WT084655MA and 102161/B/13/Z; Deutsche Forschungsgemeinschaft (SCHE 2112/1-2) and (BA 1793/6-2); the Leibniz Collaborative Excellence Programme (K622/2024), the UKRI-BBSRC Interdisciplinary Bioscience Doctoral Training Partnership (BB/M011224/1),
 - correct the nomenclature of EV Figure files to Figure EV1, etc.,
- upload EV tables as separate Expanded View Content files; to keep them in the manuscript, rename the tables as Table 1 and Table 2 and update the callouts in the text,
- remove the reagents and tools table from the main manuscript and upload as a separate file,
 - upload each figure's source data as a separate file,
 - define the annotated p values ****/**/**/* and provide the exact p-values for the same in the legend of figure EV2 B as appropriate,
 - indicate the statistical test used for data analysis in the legend of figure EV2 B,
 - define 'n' in the legends of figures 1E and 5D, and
 - define error bars in the legend of figure 5D.

We include a synopsis of the paper (see <http://emboj.embopress.org/>). Please provide me with a general summary image, a two sentence statement and 3-5 bullet points that capture the key findings of the paper.

I am looking forward to receiving your revised manuscript.

EMBO Press is an editorially independent publishing platform for the development of EMBO scientific publications.

Best wishes,

William

William Teale, PhD
Editor
The EMBO Journal
w.teale@embojournal.org

Read our guidance for manuscript revisions and related editorial policies: <https://link.springer.com/journal/44318/submission-guidelines#cms-Revised-submissions>

<https://media.springernature.com/original/springer-cms/rest/v1/content/27825798/data/v1>

- a point-by-point response to the referees' comments, with a detailed description of the changes made (as a word file).
- a word file of the manuscript text.
- individual production quality figure files (one file per figure)
- a complete author checklist
- Expanded View files (replacing Supplementary Information)
- a Reagents and Tools Table as part of the Methods section

Please remember: Digital image enhancement is acceptable practice, as long as it accurately represents the original data and conforms to community standards. If a figure has been subjected to significant electronic manipulation, this must be noted in the figure legend or in the 'Methods' section. The editors reserve the right to request original versions of figures and the original images that were used to assemble the figure.

We realize that it is difficult to revise to a specific deadline. In the interest of protecting the conceptual advance provided by the work, we recommend a revision within 3 months (14th Apr 2026). Please discuss the revision progress ahead of this time with the editor if you require more time to complete the revisions. Use the link below to submit your revision:

Referee #1:

The authors provided thorough and thoughtful responses to the reviewers' comments, therefore I recommend acceptance

Referee #2:

The revised manuscript addressed my concerns.

The underlined part of the following sentence from the third paragraph of the 'Introduction' is missing some clarifying words.

The GPCR-mediated inhibition of TASK channels has been proposed to involve many different factors though is principally via Gαq-coupled receptors.

All minor editorial requests have been addressed by the authors.

Dear Stephen,

I am pleased to inform you that your manuscript has been accepted for publication in the EMBO Journal.

Congratulations to all involved! This was a lovely mechanistic study.

You may qualify for financial assistance for your publication charges - either via a Springer Nature fully open access agreement or an EMBO initiative. Check your eligibility: <https://link.springer.com/journal/44318/how-to-publish-with-us>

Best wishes,

William

William Teale, PhD
Editor
The EMBO Journal
w.teale@embojournal.org

Please note that it is The EMBO Journal policy for the transcript of the editorial process (containing referee reports and your response letters) to be published as an online supplement to each paper. If you should prefer removal of any referee-only figures included in the point-by-point response(s), e.g. because they may still be used for future publication or because they have been reproduced from published work by others, please do let us know immediately via response email.

More information is available here: <https://link.springer.com/partners/embo-press/editorial-policies#Peer%20review>